# Bionomics of the African Apefly (*Spalgis lemolea*) as A Potential Natural Enemy of the Papaya Mealybug (*Paracoccus marginatus*) in Tanzania

**Sayuni P. Nasari** [1,*]**, Anna C. Treydte** [1]**, Patrick A. Ndakidemi** [1] **and Ernest R. Mbega** [1]

Department of Sustainable Agriculture, Biodiversity and Ecosystems Management, Nelson Mandela African Institution of Science and Technology, Arusha P. O. Box 447, Tanzania

* Correspondence: nasaris@nm-aist.ac.tz

**Abstract:** The African apefly (*Spalgis lemolea* Druce) is a potential natural enemy of the papaya mealybug (*Paracoccus marginatus* Williams and Granara de Willink). We studied the life history of apeflies in the laboratory at a temperature of 25–27 °C and a relative humidity of 55%–65% under a 12 h photoperiod condition. The papaya mealybugs and apefly larvae were collected from papaya plants in Tengeru, Arusha, Tanzania. The papaya mealybugs were introduced and allowed to multiply on potted sprouting potato plants in screened cages. In order to study the life cycle and predation of apeflies, an apefly egg was placed on an open screen-covered petri dish containing a moist blotter paper and observed for larva emergence. After the apefly larva emergence, a mixture of mealybug eggs (up to 1500), nymphs (200–250) and adults (100–150) was introduced in the petri dish each day and the consumption rate by the apefly larvae was quantified until the larvae reached pupal stage. Then, the apefly adults were collected and put into cages 30 cm × 30 cm × 30 cm containing cotton wool soaked in water, for observation of pre-mating, mating, egg-laying and life span. Results indicate that the apefly completed its life cycle in 23 days. The mean development period of the eggs was four days and the development period for the larva, pre-pupa, and pupa was nine, one and ten days respectively. The adult apefly emerged after 9 days of the pupa stage with mean body lengths of 10.1 ± 0.4 mm and 11.0 ± 0.8 mm for the males and females, respectively. The female laid an average of 68 eggs in groups of two to seven at different sites after 4–5 days of emergence. In this study one apefly larva was capable of consuming about 1983 ± 117, 123 ± 6 and 80 ± 9 papaya mealybug eggs, nymphs and adults respectively during larval stage. These results reveal the predatory potential of the apefly in suppressing the population of papaya mealybugs under field conditions.

**Keywords:** biology; natural enemy; predation; larval instars

## 1. Introduction

Fruit and vegetable crops are highly crucial for human nutrition and income in developing countries. However, farmers often experience massive yield losses due to attacks by insect pests such as the papaya mealybug (*Paracoccus margnatus* (Williams and Granara de Willink, 1992)). The papaya mealybug damages several tropical crops and weed plants worldwide [1–3], especially those belonging to the following genera: *Acacia*, Martius (1829); *Annona* L.; *Bidens*, L.; *Capsicum*, L.; *Hibiscus*, L.; *Mangifera*, L; *Manihot*, Mill.; *Persea*, Mill.; *Plumeria*, L.; *Punica* L.; *Solanum* L.; and *Vigna*, L. [4–8]. The papaya mealybug is a native to Central America and was first reported in the Caribbean and South America in the 1990s [9,10]. The pest further spread to the Pacific Islands, Bangladesh, Cambodia, Philippines and Thailand [1]. In Africa, it was first reported in Ghana in 2010, and later spread to Benin, Nigeria, Togo and Gabon [10]. In East Africa, it was reported in Tanzania in 2015 [11,12] and Kenya in 2016 [12].

Economic losses resulting from damage done by the papaya mealybug in papaya have been reported to be above economical levels [13,14]. For instance, in East Africa, the yield losses are reported to be up to 100% in papaya [12,13]. The mealybug injects a toxin as it feeds on leaves and fruit which results in yellowing of the leaves, stunting, leaf deformation, early leaf and fruit drop, and accumulation of honeydew [7,12,13]. Furthermore, sooty mold growing on honeydew emitted by the mealybugs interferes with photosynthesis resulting in a total plant death [14]. In the management of this pest, famers use synthetic insecticides such as Cypermethrin or Dimethoate [13,15,16]. However, these chemicals are not effective due to the anatomy and physiology of the papaya mealybugs' bodies, which are covered in waxy materials that make it difficult to reach them with chemicals [13–15]. Consequently, there has been increased application of pesticides [13,16] resulting in serious human health concerns [13], increased toxicity, resistance by insect pests and distortion of biodiversity [17]. In addition, the pesticides used in the control of the papaya mealybugs are usually broad spectrum and are associated with adverse effects on beneficial arthropods that would control pests biologically [14,17]. Furthermore, synthetic pesticides cause residue problems on the fruits, making them unfit for export and hazardous to domestic markets [14]. Thus, the demand for a more productive, species-specific and environmentally-friendly approach to papaya mealybug control is needed.

Different natural enemies of the papaya mealybug are commercially available including mealybug destroyer *Cryptolaemus montrouzieri* (Mulsant) (Coleoptera: Coccinellidae), lady beetles *Coccinella magnifica* (Latreille) (Coleoptera: Coccinellidae), lacewings *Chrysoperla carnea* (Stephens) (Neuroptera: Chrysopidae) and hoverflies (Diptera: Syrphidae), all of which have a potential impact on mealybug populations when uninterrupted by broad-spectrum chemicals [7,12,13]. In addition to predators, several parasitoids have the ability to attack papaya mealy bug in 95% to 100% of the cases [7]. In some countries such as the USA, parasitoids such as *Acerophagus papaya* (Noyes and Schauff), *Anagyrus loecki* (Noyes and Menezes) (Hymenotera: Encyrtidae), *Anagyrus californicus* (Compere), *Pseudleptomastix mexicana* (Noyes and Schauff), and *Pseudaphycus* sp (Hymenoptera: Encyrtidae) are used for biological control of the papaya mealybug [5,14,15].

Some Lepidopterans seem to have the potential to become successful biological control agents [16]. Members of the subfamily Miletinae feed on ant brood or Hemiptera [16], and the genus *Spalgis* Moore is known to feed on different species of mealybugs (Pseudococcidae) that are serious pests in a wide range of economically-important crops like citrus, cassava, cotton, papaya, mangoes and ornamentals [17]. Studies on the Indian apefly *Spalgis epius* Westwood (Lepidoptera: Lycaenidae) indicate that butterfly larvae can be used for the bio-control of different species of mealybugs such as *Ferrisia virgata* (Cockerell), *Paracoccus citri* (Risso), *Paecilomyces lilacinus* (Cockerell.), *P. marginatus* (Williams and Granara de Willink), and *Maconellicoccus hirsutus* (Green) (Hemiptera: Pseudococcidae), and scale insects such as *Chloropulvinaria polygonata* (Cockerell) (Hemiptera: Coccidae) and *Dactylopius* sp. (Hemiptera: Dactylopiidae) [3,18]. While previous studies have provided detailed description of the Indian apefly, little information is available on the biology or feeding behavior of the African apefly, or how it interacts with its prey [19].

The African apefly appears in Western, Eastern and Central Africa [20–23]. The literature indicates that the development, life characteristics and feeding behavior of the apefly towards mealybugs are not well known [24]. The objective of this study was, therefore, to study the development stages and predatory activity of the apefly towards papaya mealybugs in Tanzania. The findings of the study form the foundation for further exploration of the apefly's potential for biological control of the papaya mealybug.

## 2. Materials and Methods

### 2.1. Laboratory Rearing of the Prey (Papaya Mealybugs)

Papaya mealybug eggs, nymphs, and adults were initially collected using a camel hairbrush from an infested papaya plant *Carica papaya* (L.) (Brassicales: Caricaceae) in the pawpaw field of the

Tengeru Horticultural Research Institute in Arusha, Tanzania. The mealybugs were reared in the laboratory at 25–27 °C, under a relative humidity (RH) of 55%–65%, on potted sprouted Irish potatoes *Solanum tuberosum* (L.) (Solanales: Solanaceae), adopting the Technology for Production of Natural Enemies [25]. For constant availability of the prey, fresh potato sprouts were infested with papaya mealybugs whenever required.

### 2.2. Laboratory Rearing of the Predator (Apefly)

Apefly larvae were collected from mealybug-infested papaya plants at Tengeru Horticultural Research Institute Arusha, Tanzania. The larvae were reared in the laboratory on potato plants *S. tuberosum* infested with papaya mealybugs following the modified method by Dinesh et al. [26], who assessed the predation of *S. epius* on the pink hibiscus mealybug *M. hirsutus*. The predator larvae completed their development on the mealybug-infested potato plants. The emerging apefly adults were released in cages of 30 cm × 30 cm × 30 cm for mating and oviposition. Mealybug-infested potato plants were introduced inside the cage for the butterflies to lay eggs on and balls of cotton wool soaked in water for feeding. The larvae that emerged from the laid eggs were reared following the same procedure as described above. A few larvae from each instar were stored in 70% alcohol for measurement purposes [3].

### 2.3. Studies on Development and Predatory Potential of Apefly

The growth of the apefly larvae was monitored by daily investigation of the petri dishes for molting. The developmental period from egg to adult (in days) and the number of larvae instars in the life cycle were recorded. The size of the egg, each larva's instar, pre-pupa, and pupa, and the adult apefly, were all measured with an optical microscope (Optika B-350—Italy). Eclosion timing, pre-mating, mating, and egg-laying behavior of the adults, and larval feeding habits, were all recorded.

To study the potential of apefly larva as a papaya mealybug predator, quantification of prey-consumption was assessed in the laboratory at 25–27 °C, under 55%–65% RH (as described in Section 2.2.), for a 12 h photoperiod following the method by Dinesh and Venkatesha [26] with some modifications. To determine the predatory potential of the apefly on the papaya mealybug, two measures were employed: the daily and stage-wise consumptions. The consumption of papaya mealybugs by apeflies was determined using a modified method by Saengyot and Burikam [3], by assessing the daily prey-consumption of apeflies from hatching to pre-pupa. Five apefly eggs were collected with a fine camel hairbrush and kept individually in five petri dishes on filter papers to hatch. In each replicate, the emerged larva remained in the same petri dish and they were given a pre-determined amount of prey stages (i.e., eggs (up to 1500), nymphs (200–250) and adults (100–150)) for feeding until they reached the pre-pupa stage. This experiment was repeated five times, making a total of 25 eggs. Prey stages were counted using a stereo zoom microscope (ZX16/SZX10-JAPAN). The larvae excrement and unused prey (eggs, nymphs or adults) were removed daily, and the predators were fed with fresh prey every day. The amount of prey stages consumed by the apefly larvae was recorded daily.

### 2.4. Data Analysis

The independent-samples t-test was used to compare the mean body weights and wingspans of the apeflies between sexes at a 95% confidence level. The data on prey consumption were subjected to analysis of variance (ANOVA) using the STATISTICA 10th edition computer software at 5% level of significance. Treatment mean comparisons were done using the Fisher's Least Significance Difference (LSD) test.

## 3. Results

### *3.1. Morphology and Life Cycle of Spalgis Lemolea Lemolea*

#### 3.1.1. Eggs

The eggs of the apefly were laid interspersed in small groups of two to eight eggs, with many short flights in between different spots (Figure 1). The eggs were disc-shaped, flat on both sides with depressions on the tops. The eggs were cream in color, with a mean diameter of 0.51 ± 0.02 mm (Figure 1).

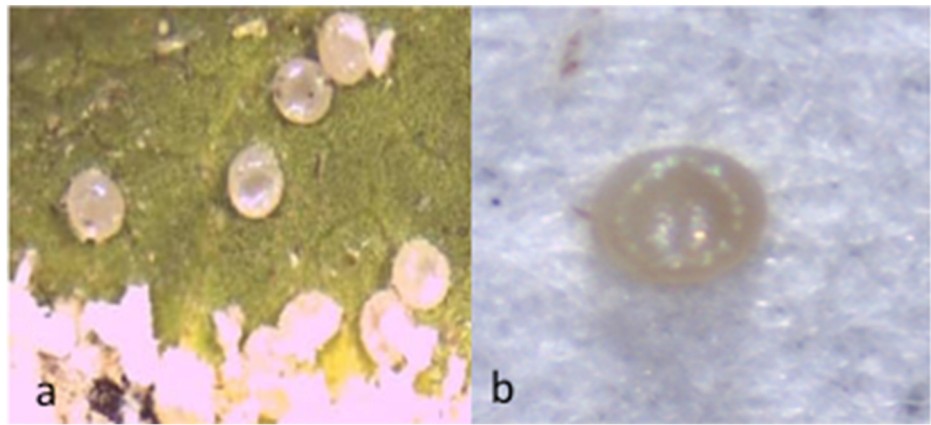

**Figure 1.** (**a**) Eggs of apefly are laid on potato leaves infested by papaya mealybugs; (**b**) An apefly egg placed on a filter paper.

#### 3.1.2. Larva

The apeflies developed through four larva stages (Figure 2). The first larval instar was sedentary, dwelling in the mealybug cluster, and made a feeding hollow in the mealybug egg masses. It also consumed small mealybug nymphs in the absence of eggs. After molting, the second larval instar moved out of the feeding chamber; its body was covered with mealybug wax and eggs. The third and fourth larval instar crawled while feeding and ate all mealybug stages. The length and width of the four larval instars are displayed in Table 1. The freshly enclosed fourth larval instar had much shorter setae compared to the previous stages, with a pale brown segmented body. All larval instars had a hard, dark brown, hairy cuticle covered with a thick waxy coating before molting (Figure 2b) as a result of its close association with mealybugs. The first and last larval instars measured 1.9 ± 0.17 mm and 10.24 ± 0.23 mm in length and 0.64 ± 0.03 mm and 6.08 ± 0.61 mm in width, respectively (Table 1).

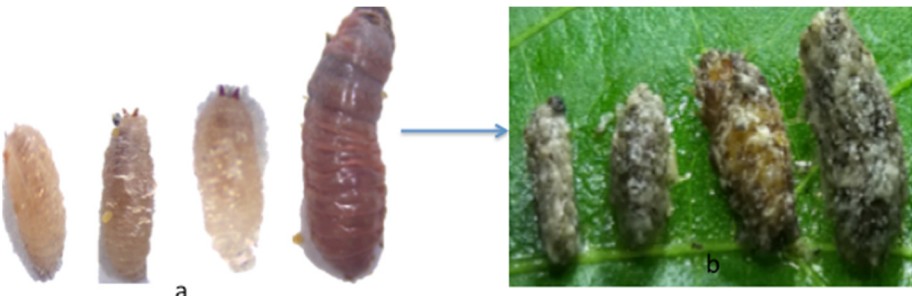

**Figure 2.** (**a**) Freshly enclosed larval instars (**b**) Larval instars before molting.

**Table 1.** Mean length and width (±SE) of apefly larva, pre-pupa and pupa stages (n = 25).

| Growth Stage | Larva Height (mm) | Larva Width (mm) |
|---|---|---|
| Instar 1 | 1.90 ± 0.03 | 0.64 ± 0.01 |
| Instar 2 | 3.27 ± 0.04 | 1.61 ± 0.02 |
| Instar 3 | 5.77 ± 0.04 | 3.67 ± 0.06 |
| Instar 4 | 10.24 ± 0.05 | 6.08 ± 0.12 |
| Pre-pupa | 7.41 ± 0.04 | 4.59 ± 0.07 |
| Pupa | 6.73 ± 0.04 | 3.94 ± 0.01 |

### 3.1.3. Pre-Pupa and Pupa

The pre-pupa larva stopped eating, shrunk and its setae disappeared (Figure 3a). It then moved away from the mealybug colony and attached itself to the leaf or stem of the host plant and produced a small amount of silk for attaching to the plant. The pupa was light brown on the dorsal-lateral and whitish on the ventral side. As development proceeded, the color darkened. The dorsal side of the pupa looks like the face of a monkey (Figure 3b).

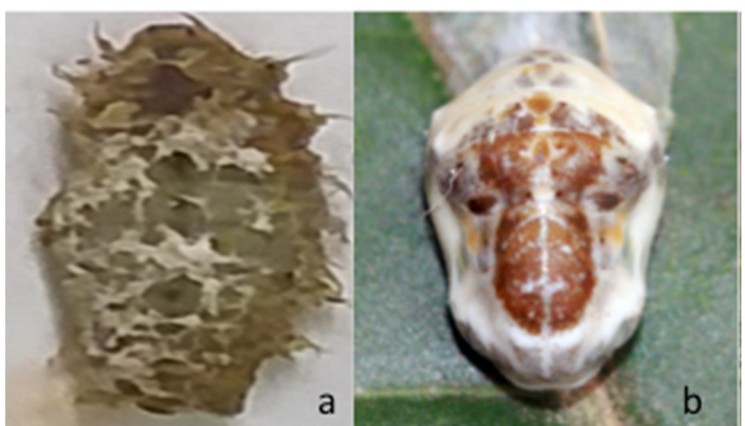

**Figure 3.** (**a**) Pre-pupa and (**b**) Pupa of apefly.

### 3.1.4. Adults

The adult apefly had whitish-grey wings with thin black strips on the inner side of the wings (Figure 4) and bold black stripes on the outer edges of the fore wings. There were no significant color differences between the male and female adults the abdomen of the male was slim but wider in females, the former with an ovipositor for egg-laying. Observations of the pre-mating behavior revealed a prolonged physical contact whereby the females pushed underneath males to mate. The mean body lengths of the male and female adults were 10.10 ± 0.43 mm and 11.03 ± 0.84 mm respectively, and the average wingspans were 27.15 ± 0.65 mm for males and 29.76 ± 1.01 mm for females. There was a significant difference in body lengths (t = −4.384, $p < 0.001$) and wingspans (t = −9.684, $p < 0.001$) between the males and females. The female laid two to seven eggs in a group at different spots, making an average of 68 eggs after 4–5 days of eclosion.

The duration of each developmental stage of *S. lemolea* was determined under laboratory conditions at 25–27 °C, 55%–65% RH and with a 12 h L:12 h photoperiod.

The mean duration of the development of eggs, larva instars, pre-pupa and pupa were 3.7, 10, 0.95, and 9.48 days respectively, as shown in Table 2. The life span of the apeflies from the egg to adult emergence was 23.5 days.

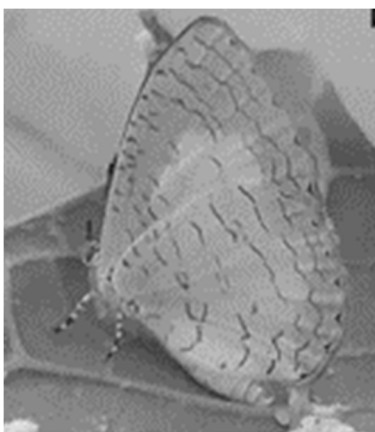

**Figure 4.** Adult female of apefly laying eggs.

**Table 2.** Duration of developmental stages of apefly reared on papaya mealybugs under laboratory conditions (n = 25).

| Developmental Stage | Duration (Days) | | |
|:---:|:---:|:---:|:---:|
| | Mean ± SD | Min–Max | Median |
| Egg | 3.66 ± 0.41 | 3.11–4.50 | 3.60 |
| Larva 1st Instar | 2.27 ± 0.60 | 1.50–3.91 | 2.21 |
| 2nd Instar | 2.06 ± 0.29 | 1.72–2.60 | 2.02 |
| 3rd Instar | 2.01 ± 0.29 | 1.60–2.53 | 1.90 |
| 4th Instar | 3.30 ± 0.51 | 2.41–4.20 | 3.31 |
| Pre-pupa | 0.95 ± 0.16 | 0.72–1.61 | 0.92 |
| Pupa | 9.48 ± 1.19 | 8.01–12.00 | 9.01 |

*3.2. The Predatory Potential of Apefly*

Four larval instars completed their developmental stages in 10 days in the laboratory conditions. First-instar larvae consumed a mean of 77.4 ± 6.5 eggs of papaya mealybugs in the first day, showing an increase in consumption in the following stages. The highest consumption of 311.2 ± 20.3 eggs of papaya mealybugs was reached on the ninth day but then decreased to 288.8 ± 19.5 on the 10th day (Table 3).

The consumption of papaya mealybug nymphs by a newly-hatched apefly on the first day was 1.2 ± 0.4 nymphs, increasing as the development progressed. The maximum level of nymph consumption, 24.6 ± 1.1, was reached on the ninth day but decreased to 20.8 ± 1.1 on the 10th day. When papaya mealybug adults were provided as prey, the consumption by a newly-hatched apefly larva was 0.2 ± 0.4 adults on the first day, but increased after that to 16.6 ± 1.8 adults on the ninth day and then decreased to 14.8 ± 2.8 on the 10th day (Table 3).

On average, a single apefly larva consumed 1982.6 ± 117 eggs, 123 ± 5.8 nymphs and 80 ± 8.5 adult papaya mealybugs during its entire larval development. The consumption of prey increased as the apefly progressed in development.

The consumption of papaya mealybugs by the apefly larval instars differed in the mean number of prey eggs, nymphs and adults consumed by the different instars. The first-instar larvae of the apefly were almost stationary while feeding, but the second-, third- and fourth-instar larvae crawled as they fed. In general, instar-wise, prey consumption showed that the older the apefly larvae the higher the number of prey consumed.

**Table 3.** Daily prey consumption of papaya mealybugs in different life stages by apefly larva.

| Instar Stage | Days | Egg | % Increase | Nymph | % Increase | Adult | % Increase | F Statistics | *p* Value |
|---|---|---|---|---|---|---|---|---|---|
| Instar 1 | 1 | 82.2 ± 1.32 a | | 2.2 ± 0.37 b | | 1.6 ± 0.24 b | | 3323.9 *** | $p \leq 0.001$ |
| Instar 1 | 2 | 171.6 ± 4.41 a | 109 | 3.6 ± 0.51 b | 64 | 3.2 ± 0.20 b | 100 | 1431.75 *** | $p \leq 0.001$ |
| Instar 2 | 3 | 262 ± 6.49 a | 53 | 6.6 ± 0.68 b | 83 | 4.6 ± 0.24 b | 44 | 1542.56 *** | $p \leq 0.001$ |
| Instar 2 | 4 | 353.8 ± 8.97 a | 35 | 17 ± 0.84 b | 158 | 10.8 ± 0.58 b | 135 | 1416.53 *** | $p \leq 0.001$ |
| Instar 3 | 5 | 436.6 ± 6.44 a | 23 | 21.4 ± 1.17 b | 26 | 15.4 ± 0.87 b | 43 | 4013.73 *** | $p \leq 0.001$ |
| Instar 3 | 6 | 817 ± 26.86 a | 87 | 36.6 ± 1.96b | 71 | 19.8 ± 1.24 b | 29 | 856.14 *** | $p \leq 0.001$ |
| Instar 3 | 7 | 950.2 ± 19.65 a | 16 | 47.2 ± 1.11 b | 29 | 24.8 ± 1.59 b | 25 | 2144.93 *** | $p \leq 0.001$ |
| Instar 4 | 8 | 1220.8 ± 30.76 a | 28 | 80.2 ± 4.22 b | 70 | 32.8 ± 1.32 b | 32 | 1405.46 *** | $p \leq 0.001$ |
| Instar 4 | 9 | 1273.4 ± 19.06 a | 4 | 95.8 ± 2.42 b | 19 | 38.4 ± 2.50 c | 17 | 3881.22 *** | $p \leq 0.001$ |
| Instar 4 | 10 | 1221.6 ± 12.76 a | −4 | 86 ± 4.35 b | -10 | 30.8 ± 1.39 c | −20 | 7377.9 *** | $p \leq 0.001$ |

Values presented are means ± SE. Different letters within the same row are significantly different at *p* = 0.05 as determined by Fisher's Least Significance Difference test. *** = Significant at $p \leq 0.001$.

## 4. Discussion

The present study aimed to describe the morphology, life cycle and the predatory activity of apeflies. The findings revealed that apefly eggs took a mean duration of 3.6 days until larva emergence, after which the larvae underwent four larval instars, the pre-pupa stage and pupa stage to reach the adult stage. These findings are consistent with other carnivorous lycaenid butterflies *Feniseca tarquinius* (Fabricius) and *S. epius*, as reported by Hall et al. [27] and Dinesh and Venkatesha [18] respectively, and other phytophagous lycaenids like *Rapala takasagonis* (Matsumura) [28] and *Lampides boeticus* L. L. [29]. However, the four larval instars of the apefly were comparatively similar to those reported in other species of lycaenids such as *S. epius* [26], *P. pyrodiscusblucida* [30] and *Lycaeides melissa samuelis* (Nabokov) [31]. Consistent with Lamborn [32], the first-instar larva of the apefly was mostly stationary while other instars were mobile and covered with a white waxy material which camouflaged them with the mealybugs. Similar findings were reported in species such as *S. substrigata* (Snell) in the Philippines [33], *S. epius* in India [3,18,26] and *Feniseca Tarquinius* (Fabricius, 1793) [30]. The mean duration for all four instars was 9.7 days, representing feeding days that can significantly bring down pest population.

Moreover, the present study showed that the first-instar larvae mostly fed on the eggs of mealybugs, while the second instar fed on its eggs and young nymphs and the third and fourth instars fed intensively on all stages. However, all four larval instars of the apefly consumed more eggs than nymphs and adults. This could be due to their small size compared to the nymph and adult stages of papaya mealybugs, as reported in the predation of *P. citri* by *S. epius* [26], and *M. hirsutus* by *C. montrouzieri* [34]. The daily instar-related prey consumption of papaya mealybugs by apeflies increased as they progressed in development. Similar findings are reported on *S. epius* regarding *M. hirsutus* consumption [18,34]. However, the consumption of eggs was higher than in other stages as reported by Saengyot and Burikam [3] for *S. epius*, which might be associated with the inability of the prey to escape from the apefly larvae. On average, a single apefly larva consumed 1982.6 ± 117 eggs, 123 ± 5.8 nymphs and 80 ± 8.5 adult papaya mealybugs during its entire larval development period. This consumption is high compared to the findings on *S. epius* by Saengyot and Burikam [3], which showed that the total numbers of prey consumed during the larval instars by the four larval stages were 4115.75 ± 553.28 eggs, 281.25 ± 45.08 nymphs and 77.50 ± 16.52 adults. These findings reveal the predatory potential held by the apefly, and indicate the necessity of further research on its efficacy under field conditions and how it can be adopted better for integrated pest management strategies for mealybug pests.

The mean total larva period in an apefly was 10 days, whereas in the predatory *S. epius* in India it was 9.4 days [26] and 11.9 days in *L. boeticus* [29]. It is reported that aphytophagous lycaenid larvae do not spend as much time as larvae as the phytophagous lycaenids [35,36]. The monkey-faced pupa was similar to that of *S. epius* and *F. tarquinius* [27]. Studies show that the monkey resemblance for some lycaenid pupae is useful in self-defense [37]. The butterflies emerged randomly from the pupae of the same age in the laboratory. The external morphology of apefly adults was similar to that of *S. epius* [26] except for the color patterns. The intermittent flight and egg-laying pattern demonstrated by the apefly have also been reported in *C. xami* by Cordero et al. [38]. The average duration of an apefly adult was 9 to 13 days from emergence. The observed pre-mating behavior included prolonged antenna contact and physical contact, wherein the females pushed underneath the males to mate, as supported by Myers [39]. Oviposition by the adult females was witnessed near mealybug colonies. The eggs of the apefly were creamy in color and disk-shaped, and both their depressed top and flattened bottom are also reported in the predatory lycaenid *S. epius* [18] and *Feniseca tarquinius* [27], and phytophagous lycaenids such as *Paralucia pyrodiscus lucida* Cros [30], *R. takasagonis* Matsumura [28] and *L. boeticus*. [29].

## 5. Conclusions

This study provides detailed information on the biology and feeding behavior of apeflies as compared to related species in Asia. A single apefly larva preyed effectively on a large number of

papaya mealybugs. From this investigation, the predatory potential of African apeflies as a biological control agent of papaya mealybugs was evident. These findings recommend further research on the apefly's efficacy in suppressing the populations of papaya mealybugs under field conditions. Moreover, the study of the population dynamics of the apefly and the papaya mealybug is warranted in order to support the integration of the apefly in papaya mealybug management strategies. Furthermore, assessment of the presence of other native natural enemies of the papaya mealybug and evaluation of their capacities would be helpful in supporting the biological pest control efforts in Tanzania and Sub-Sahara Africa.

**Author Contributions:** Conceptualization, S.P.N.; methodology, S.P.N, E.R.M. and P.A.N.; Field investigations and data collector, S.P.N; writing—original draft preparation, S.P.N.; writing—review and editing, E.R.M, A.C.T. and P.A.N.; supervision, E.R.M, A.C.T. and P.A.N.; project administration, S.P.N.; funding acquisition, S.P.N. All authors have read and agreed to the published version of the manuscript.

**Funding:** This research was funded by the AFRICAN DEVELOPMENT BANK, Grant number 2100155032816

**Conflicts of Interest:** The authors declare no conflict of interest. The funders had no role in the design of the study; in the collection, analyses, or interpretation of data; in the writing of the manuscript, or in the decision to publish the results.

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
