# Peer review of "Bionomics of the African Apefly (Spalgis lemolea) as A Potential Natural Enemy of the Papaya Mealybug (Paracoccus marginatus) in Tanzania"

_sustainability, doi:10.3390/su12083155_

Round 1

Reviewer 1 Report

The overall presentation of the paper could be improved significantly. Writing style must be improved. I have provided some suggestions for improvement, but rewriting entire sections is beyond the scope of this review. 

There are numerous grammatical and typographical errors throughout the manuscript, which could have easily been corrected with a more careful proof reading and reading other published work, to see how information is presented.  A number of technical errors (e.g., in spellings of scientific names) were also seen, which is really not acceptable in a scientific paper. 

The study is simple and straightforward. It uses standard methodology, and the results have been explained suitably.  

Pictures are not clear, and this makes it difficult to understand features being pointed out. 

Your study is a good piece of information that must reach the concerned audience.  Make sure it does!

Please take some time and re-visit your paper. 

Read several published articles in peer-reviewed journals, which address topics similar to yours, and others as well.  Observe how those papers are written and pay attention to how they are presented.

Rewrite the entire paper with those papers in mind, and paying careful attention to reviewers’ comments on how you can improve the manuscript.  

Author Response

Dear reviewer 1, we thank you so much for your constructive comments that has improved our manuscript significantly.

We have revised the entire manuscript and your comments have been implimented.

Please note that the line numbers have changed due to the improvement of the introductory part, therefore line numbers that appears in your comments may not be the same in this new document.

Please find the attached document for further details on implementation of the comments.

Best regards,

Reviewer 2 Report

General comments. The manuscript "Bionomics of the Apelfy (Spalgis lemolea lemolea) as  potential natural enemy of the Papaya mealy bug  (Paracoccus marginatus Williams and Granara de Willink) in Tanzania” describe the biology of this rare buttefly that can be preserved and used as candidate for biological control of the Papaya mealy bug. The manuscript gives biological information of value. However in many aspect this is rather descriptive since many relationships, e.g. between the consumption rate and larva development should have been tested by the appropriate statistics. Moreover few points need to be sort out in order to reconsider again this manuscript for publication. My main concern is about the reliability of the experiment. In fact authors do not specify exactly the instars nymph stage of the mealybug that was attacked daily by the butterly larvae. As the size of the nymphs can change with their stage, I think that this aspect can influence the data obtained. Finally, the introduction is often too brief and not gives enough information about the malybug damage, distribution etc. Consequently the objectives are not introduced appropriately. Sometimes English form is unclear and I suggest deep revision by a mother tongue.

in detail:

Line 22-23. The authors should point out why this study allows further research on the efficacy of S. lemolea in suppressing the population of P. marginatus. In reality there is only a description of on the biology and feeding behavior of African apefly.

Introduction: the authors should describe more in detail the pest, in particular: the distribution, the type of damage, the host plants, the economical losses determined if data are available,  etc…

Line 29 add latin name of papaya mealybug

Line 31-34 this part is a little confusing, shlod be rewritten. More in detail, which are the common insecticides used to control this pest?

Lines 40-45. This part should also be rewritten, is this species a predator? In which way it attacks dangerous phytophagous pest?

Line 63. Add descriptor of Carica papaya

Line 80-87. Were the author using some recording system or some software for the observation for the behavior (eg observer)? If not they should point out how the observation were carried out.

Line 98-101. This experiment is hardly replicable. In fact the authors write that “the larva was kept in the same petri dish and provided with a daily known number of egg masses, nymphs and adults…” but than is not easy to understand if each replicate was conducted in the same way or not. So I advise to rewrite this part making this clearer.

Line 100-102. Why this was done?

Figure 1 caption. please check. Maybe the authors should put letter A,B,C to identify the different figures…

Line 178. The authors write about “The consumption of nymphs of P. marginatus” is difficult to understand if they were 1,2,3,4 instar nymphs, because this influence very much the data presented.

Line 201. Change F. tarquinius  in Feniseca tarquinius (Fabricius)

Line 202. R. takasagonis does not goes abbreviated, while in line 240 need to be abbreviated.

Line 203. L. boeticus not abbreviated here, and abbreviated later on (line 240)

Line 244-245. As last sentence I would expect something better about the effective possibility to use this agent as biocontrol candidate, please rewrite.

Author Response

Dear reviewer 2, we thank you so much for your constructive comments that has improved our manuscript significantly.

We have revised the entire manuscript and your comments have been implimented.

Please note that the line numbers have changed due to the improvement of the introductory part, therefore line numbers that appears in your comments may not be the same in this new document.

Please find the attached document for further details on implementation of the comments.

Best regards,

Round 2

Reviewer 2 Report

The paper Bionomics of the African apefly (Spalgis lemolea) as a potential natural enemy of the Papaya mealybug  (Paracoccus marginatus) in Tanzania is about the biology of a predaceous Lepidoptera of an important pest. Although the paper has been improved in some section as introduction, I don’think is suitable for publication in sustainability. The information given have some value but they are reported badly, there are few mistakes in the figure, tables are sometimes redundant and references are sloppy (many reference written in wrong way, other in very low ranked journals, etc.).Moreover also in consideration that the paper is rather descriptive, often without the adequate level of accuracy for an high-impact journal as Sustainability, and the English level is still poor, I would recommend to resubmit it to another low-ranked entomology journal after sorting out a series of corrections listed below.

line 11. Put an introductive sentence

lines 14-16. Summarize these sentences

line 16.  “To understand the” change in ” In order to study the life cycle of ….”

Line 17 delete “five replications”

Line 21. Put a space after cm

Liner 27-29 please rewrite

Line 37. Something wrong here

Line 40. Something is missing

Line 41. Caribbean is misspelled

Line 44. Delete )

Line 46-62 in general this part need to be summarized.

Line 48. 100% of what?

Line 49. “use synthetic chemicals such as those with Cypermethrin and Dimethoate…” change in “use synthetic insecticides as cypermethrin or dimethoate …” also add “However…

Line 51. “Moreover the papaya mealybugs…”

Line 63-70 summarize and please add all the species descriptors!

Line 67 “parasitoids have been tried and confirmed to have the ability to attack” bad English please rewrite

Line 71-81, this part is not well linked with the rest; Line 72 add space after 20

Line 84. Add dot

Line 90-92 please add descriptors of the species or abbreviate if already reported!

Line 115-128. Improve english and correct several mistakes about spaces and capital letter

Figure 1 is not convincing me, is the same picture with different enlargements? There is not scale, no letter

Figure 2, what is i, ii, iii, and iv? Moreover here is specificed that photographs is from the author. Then the others are not? In that case the authors have to indicate the source

Table 1. what are these numbers!? cm? It is really difficult to understand. Personally I would remove this table. Does not have meaning say that instar 1 is bigger than instar 4

Line 191. I dont understand. Only four individual complete the cycle? If these are the total numbers I don’t think the data are representative.

Figure 5. Mistake in the figure, instar is misspelled twice L

Line 215. “sought to describe” please change, this is supposed to be a scientific paper.

Line 216. Change comma with dot after “apefly”

Line 215-221. This is totally to rewrite, please underline the main findings, avoiding sentences withour any accuracy. The color of the larvae or adult is not one of the main findings. Just focus on the duration of the stages and the predator activity.

Line 228. bad English. Change in “Moreover the present study showed that the first instar larvae…”

Reference section is totally to check one by one because is full of mistakes, some ref are very old or written in journals not indexed.

Author Response

Dear Reviewer,

We thank you for your second round comments, Kindly please find the attached responses to the comments

Round 3

Reviewer 2 Report

Line 49-50. Please give information about the type of damages direct, typical of pierce-sucking insects that determine direct and indirect damages on the plants

See for example

Guarino, S., Peri, E., Colazza, S., Luchi, N., Michelozzi, M., & Loreto, F. (2017). Impact of the invasive painted bug Bagrada hilaris on physiological traits of its host Brassica oleracea var botrytis. Arthropod-Plant Interactions11(5), 649-658.)

da Silva, V. P., Galzer, E. C. W., Malausa, T., Germain, J. F., Kaydan, M. B., & Botton, M. (2016). The vine mealybug Planococcus ficus (Signoret)(Hemiptera: Pseudococcidae) damaging vineyards in Brazil. Neotropical entomology45(4), 449-451.

Line 79-80 please specify the names of the descriptors of the specie cited, moreover “Mexicana” should be not in capital letter

Line 94 and 98, add the descriptors names of the species

Line 11-112, please rewrite, “observation carried out on Petri dishes…”

Line 147, in my opinion better to delet “phtorgarhs by the author here and elsewhere in the manuscript, it is supposed you did it yourself.

Figure 4. why is written in italic?

Line 189. I continue not to understand the table 1. At least I don’t think is suitable the statistical analysis here. Just report the means ± se, but delete the ANOVA because in my opinion don’t have any meaning here. Larva moult and grow, so of course instar 4 is longer and larger than instar 1.

Table 3. there a misspelling of instar 2 at the 4th row.

Figure 5.the figure is to improve, is also missing the SE values. However, is this reporting the same data of table 3? In that case maybe better to delete it.

Line 235. Melissa not in capital letter

Line 247. Planococcus needs to be abbreviated and delete Risso

Line 274. Feniseca tarquinius, add descriptor species

Line 275. Delete Matsumura

please check that the reference style is according to the journal editing.

Author Response

Dear reviewer,

Kindly please find the attached document.
